# Genetic Characterization of the Local Pirenaica Cattle for Parentage and Traceability Purposes

**DOI:** 10.3390/ani10091584

**Published:** 2020-09-05

**Authors:** David Gamarra, Masaaki Taniguchi, Noelia Aldai, Aisaku Arakawa, Andres Lopez-Oceja, Marian M. de Pancorbo

**Affiliations:** 1Biomics Research Group, Lascaray Research Center, University of the Basque Country (UPV/EHU), 01006 Vitoria-Gasteiz, Spain; davidgamarrafdz@gmail.com (D.G.); andreslopezoceja@gmail.com (A.L.-O.); 2Animal Genome Unit, Institute of Livestock and Grassland Science, National Agriculture and Food Research Organization (NARO), Tsukuba 305-0901, Japan; masaakit@affrc.go.jp (M.T.); aisaku@affrc.go.jp (A.A.); 3Lactiker Research Group, Department of Pharmacy and Food Sciences, University of the Basque Country (UPV/EHU), 01006 Vitoria-Gasteiz, Spain; noelia.aldai@ehu.eus

**Keywords:** structure, identity, assignment test, microsatellite, multiplex PCR, Salers, Holstein-Friesian, Terreña, Blonde d´Aquitaine, Limousin

## Abstract

**Simple Summary:**

Domestic livestock diversity is an important component of global biodiversity and molecular data have become essential for the characterization of genetic diversity in cattle. The aim of this study was to assess the effectiveness of a 30-short tandem repeat (STR) panel and reveal the genetic structure of a local Pirenaica breed compared with other breeds (Terreña, Blonde d’Aquitaine, Limousin, Salers and Holstein-Friesian) typically raised in the same geographic Basque region. The proposed STR panel could be used as an appropriate genetic tool to trace Pirenaica animals and their Protected Geographic Indication (PGI) products.

**Abstract:**

Pirenaica is the most important autochthonous cattle breed within the Protected Geographic Indication (PGI) beef quality label in the Basque region, in northern Spain. The short tandem repeats (STRs) are powerful markers to elucidate forensic cases and traceability across the agri-food sector. The main objective of the present work was to study the phylogenetic relationships of Pirenaica cattle and other breeds typically raised in the region and provide the minimum number of STR markers for parentage and traceability purposes. The 30-STR panel recommended by the International Society of Animal Genetics-Food and Agriculture Organization of the United Nations (ISAG-FAO) was compared against other commercial STR panels. The 30-STR panel showed a combined matching probability of 1.89 × 10^−25^ and a power of exclusion for duos of 0.99998. However, commercial STR panels showed a limited efficiency for a reliable parentage analysis in Pirenaica, and at least a 21-STR panel is needed to reach a power of exclusion of 0.9999. Machine-learning analysis also demonstrated a 95% accuracy in assignments selecting the markers with the highest F_ST_ in Pirenaica individuals. Overall, the present study shows the genetic characterization of Pirenaica and its phylogeny compared with other breeds typically raised in the Basque region. Finally, a 21-STR panel with the highest F_ST_ markers is proposed for a confident parentage analysis and high traceability.

## 1. Introduction

Pirenaica are the most important beef cattle raised in the Basque region, northern Spain, and their meat is included, together with other breeds, within the local Protected Geographic Indication (PGI) label [1]. The regulation of the PGI quality label establishes the production system and defines health and welfare requirements. This beef is highly appreciated by the local consumer, and has been registered as an “100% autochthonous breed” by the Spanish Ministry of Agriculture under the National Regulation (R.D 2129/2008, December 26th) on conservation, improvement and promotion of animal breeds [2]. Tool-supported traceability is of high importance to avoid food fraud and maintain high standards across the agri-food industry. In this sense, DNA-based traceability constitutes a powerful tool for parentage detection, and individual or species identification in the food chain [3,4]. For instance, microsatellite, or short tandem repeat (STR) loci, have become necessary tools for pedigree recording and inbreeding control and are essential for conservation and selection programs [5]. STR loci are also interesting for individual animal identification purposes through the meat chain promoting food safety and traceability [6,7].

In this sense, several STR panels have been proposed for parentage verifications in the breeding industry and cattle related forensic cases. However, depending on the studied breed, a selection of an appropriate STR panel with a minimum number of markers allowing a kinship exclusion probability over 99.99 is required. In this sense, initially, a panel of 9-STR was recommended suggesting the increase of markers in parentage testing [8], and two markers were included in a commercial 11-STR panel (StockMarks^®^, Applied Biosystems, Foster City, CA, USA). Later, a 12-STR core panel was proposed [9] and commercialized (Bovine Genotype™ Panel 1.2, Thermo Scientific, Waltham, MA, USA). Finally, six additional markers were intended (Bovine Genotype™ Panel 2.2, Thermo Scientific, USA) to be used as a complement to Panel 1.2 when more STR loci are required. For cattle genetic resources management studies, on the other hand, FAO proposed a 30-STR panel [10]. The 12-STR core panel has been widely used in parentage and identity testing, but after the poor discrimination power observed among several European breeds it was suggested to increase the amount of genotyped STRs for forensic cases [11]. In the Pirenaica breed, a 10-STR panel was studied for traceability purposes [12] and other few and/or not core STRs were reported for genetic diversity [13,14]. In other breeds included in the Basque PGI label, such as autochthonous Terreña, or imported Blonde d´Aquitaine, Limousin and Salers, genetic diversity studies have been reported [15,16,17]. However, most of previous studies were performed before the recommendation by the FAO for using a 30-STR panel, and therefore, limitations of markers make for a difficult the comparison with data from later studies [10]. Therefore, the objectives of the present study were (1) to study the genetic diversity and phylogeny of Pirenaica cattle; (2) to assess the effectiveness of the 30-STR panel, from the International Society of Animal Genetics-Food and Agriculture Organization of the United Nations (ISAG-FAO) Advisory group on Animal Genetic Diversity, for parentage and individual identification compared to other commercial STR panels recognized as the minimum standard for identity and kinship testing; and (3) to assess the traceability through genetic assignments comparing several STR panels in a single predictive model and select the most discriminative markers to determine the population of origin of an unknown individual in Pirenaica and other breeds raised in the Basque region (Terreña, Blonde d’Aquitaine, Limousin, Salers and Holstein-Friesian).

## 2. Materials and Methods

### 2.1. Sample Collection and DNA Extraction

Muscle samples from Pirenaica beef cattle from several farms located in the Basque Country (northern Spain) were collected (*n* = 114) according to the Bovine Identification Document. Neither parentage nor maternal half-sibs were observed, and paternal half-sibs were maintained at low frequencies (0.009). Furthermore, muscle samples from Salers (*n* = 13) and Holstein-Friesian cattle (*n* = 21) were collected for assignment accuracy purposes [18]. Salers and Holstein-Friesian (as cull dairy cows) are both integrated in the Basque beef supply chain. Pirenaica, Salers and Holstein-Friesian were purebred and, in all cases, neck (*Sternomandibularis)* muscle samples were collected at 24 h *postmortem* in a local commercial abattoir (Harakai Urkaiko S. Coop., Zestoa, Gipuzkoa). DNA was extracted from 20 mg of muscle tissue using a salting-out method. The DNA pellet was re-hydrated with 200 µL of H_2_O and aliquots were stored at −20 °C. DNA samples were quantified with a NanoDrop ND-1000 spectrophotometer (Thermo Scientific, Wilmington, DE, USA) and diluted to 50 ng/µL.

### 2.2. Sample Genotyping

The PCR amplification of the recommended 30 microsatellites (BM1824, BM2113, ETH03, ETH010, ETH225, INRA023, SPS115, TGLA53, TGLA122, TGLA126, TGLA227, BM1818, CSSM66, CSRM60, ILSTS006, HAUT27, HEL01, INRA005, INRA037, INRA063, ETH152, HEL09, ETH185, HEL05, HEL13, INRA032, MM12, HAUT024, ILST005 and INRA035) was performed by multiplex PCR reaction [10]. Capillary electrophoresis was performed on an ABI PRISM 3130xl Genetic Analyzer (Applied Biosystems, Foster City, CA, USA) using internal size standard GS LIZ500 (Applied Biosystems). GeneMapper v4.0 was used for fragment analysis.

### 2.3. Statistical Analysis

In order to study the genetic variation, Genepop 4.2 software [19] was used to test for deviations from Hardy–Weinberg equilibrium (HW) using the test reported by Guo and Thompson [20] and a Markov chain (dememorization 5000, batches 100, iterations per batch 1000). Bonferroni’s procedure was applied to correct the level of significance of multiple tests. The cervus 3.0.3 software [21] was used to calculate the number of alleles per locus (k), observed heterozygosity (Ho) and expected heterozygosity (He).

Genetic relationships of Pirenaica were analyzed against Terreña, Salers and Holstein-Friesian STR data from our previous studies [15,16]. Reynolds genetic distance measures were computed by Arlequin 3.5 [22]. A factorial components analysis (FCA) was used to represent a three-dimensional plot [23], and a Neighbor-joining (NJ) phylogenetic tree based on individuals was also constructed [24].

For individual identification and traceability, matching probability (MP) was computed as the probability to have a match between two individuals sharing an identical genotype profile and chosen at random [25] and using Powerstats 1.2 (Promega, Madison, WI, USA). The combined matching probability (CMP) was computed with the formula: CMP = (MP_1_)(MP_2_)…(MP_k_) which is the overall MP including *k* number of loci. For parentage purposes, power of exclusion for each locus (PE) was calculated in the absence of genetic information from one parent (PE-1P) and power of exclusion when genetic information of both parents was available (PE-2P), power of exclusion for identity of two siblings (PE-SI) was also computed. Finally, combined power of exclusion (CPE) with the formula: CPE = 1 − (1 − PE*_1_*)(1 − PE*_2_*)…(1 − PE*_k_*) which is the overall PE including *k* number of loci was calculated [26]. CPE was calculated for PE-1P (CPE_1_), PE-2P (CPE_2_) and PE-SI (CPE_SI_) using several commercial marker panels and compared with the improved discrimination power of the 30-STR panel.

Assignment tests were performed including several breeds typically raised in the Basque region. On one hand, minority PGI breeds such as native Terreña [15], Blonde d´Aquitaine and Limousin [11] were included using published STR data. On the other hand, Salers and Holstein-Friesian breeds, genotyped in this study using the 30-STR panel, were included. First of all, the assignment of individuals to their breeds was tested using the frequency-based [27] and the Bayesian-based [28] methods implemented in geneclass 2 software [29]. The Bayesian method was computed by simulating 1000 genotypes (using allele frequencies) and a fixed threshold of 0.001. Thus, an individual was considered as correctly assigned to a population when it was excluded from all of the non-origin populations (*p* ≤ 0.001), but not from the true population of origin. Secondly, the existence of distinct genetic populations and assignment was tested with a Bayesian model based method in structure 2.3.4 software [30] under an admixture model for clusters (K) and using 10 Markov Chain Monte Carlo simulations consisting of 1 × 10^5^ iterations after a burn-in of 5 × 10^5^ iterations. The optimal value of K was selected following the clustering mode described by Kopelman et al. [31] and the approach ΔK of Evanno et al. [32]. Thereafter, a supervised machine-learning approach was used to estimate the mean and variance of assignment accuracy by a Monte-Carlo resampling (100 iterations) cross-validation procedure using r software and AssignPOP package [33]. Analyses were adjusted by the proportion of individuals and by the STRs with the highest F_ST_ to estimate the minimal number of markers for an accurate assignment. This approach creates randomly selected, independent training and test data panels which avoids introducing high-grading bias [34], while the proportion of individuals from each source population randomly allocated to the baseline data panel was adjusted to avoid biases associated with unbalanced population sizes [35].

## 3. Results and Discussion

### 3.1. Genetic Variations and Genetic Relationships

A total of 30 STR loci were analyzed in 114 Pirenaica individuals and HW equilibrium was reached in all markers when Bonferroni’s correction was applied, except for the locus INRA037 (Table 1). The He across all markers varied from 0.372 (ILST005) to 0.865 (TGLA227), where the average He was 0.680. All the loci were polymorphic and 211 alleles were detected. The number of alleles per locus ranged between 2 (ILST005) and 12 (CSSM66) with an average of 7.03 ± 2.51.

The origin of Iberian breeds occurred through arrival of cattle from the *Bos taurus* (Taurine) lineage. However, MacHugh [36] observed African zebu (*Bos indicus*) diagnostic alleles in European and African taurine (*Bos taurus*) breeds, which indicates a zebu gene introgression into taurine breeds. In this pirenaica (taurine) study, the locus BM2113 showed the zebu diagnostic 131-bp allele with a frequency of 0.149. In addition, other zebu diagnostic alleles were observed at very low frequencies such as ETH152-193 (0.075) and BM2113-123 (0.009). Zebu and African-type STR alleles have previously been reported in Iberian cattle [37,38], while the African mitochondrial T1 haplogroup was also observed in Pirenaica [39]. Therefore, zebu markers may suggest a North African genetic signature in Pirenaica cattle, based on the hypothesis of Neolithic dispersal through the Mediterranean route and historical migrations [40,41].

The average heterozygosity value (0.680) was similar to the value reported in Pirenaica by Rendo et al. [14] using 11 STRs (0.688) and higher than the value reported by Cañon et al. [13] using 16 STRs (0.628). These two studies had smaller sample sizes. In contrast, Martin-Burriel et al. [42] showed the lowest He in Pirenaica, probably due to sampling or panel selection even 30 STR were genotyped. Overall, average heterozygosity value was slightly higher than values reported in native breeds from Spain, Portugal and France ranging from 0.50 to 0.71 [6,13].

The PIC values per locus varied between 0.302 and 0.845, with a mean value of 0.637. A PIC value exceeding 0.5 indicates highly polymorphic microsatellite marker, while values ranging from 0.25 to 0.5 indicate medium-polymorphic loci. In Pirenaica, the PIC values of most loci exceeded 0.6, except for HEL13 (0.421), INRA035 (0.384) and ILST005 (0.302). Based on heterozygosity and PIC values, Pirenaica had slightly lower genetic diversity than other Basque breeds such as Terreña, Monchina and Betizu [14,15]. According to Mendizabal et al. [43], the Pirenaica population increased significantly during the 1850s, but the later introgression of new cattle breeds from Europe lead to an endangered situation of the former breed. It was not until 1975, when the need to maintain sustainable production systems using native animal genetic resources promoted the recovery and improvement of Pirenaica. At present, Pirenaica has the largest population size compared to other aforementioned native Basque breeds and it is the first native cattle breed being included in a selection program in the Basque region. Therefore, the selection from a reduced number of reproducers might imply lower heterozygosity and a reduction of the number of alleles in comparison with aforementioned native breeds that have been kept in a semi wild natural environment. In fact, this study has showed a smaller mean number of alleles per locus (7.54 ± 2.5) in Pirenaica compared to Betizu (7.91 ± 2.51) or Terreña (8.64 ± 2.54) from a previous study [14]. However, our Pirenaica sample showed a higher mean number of alleles per locus than others, which ranged between 6.22 and 6.91 [12,14].

The relationship between Pirenaica and other breeds produced in the Basque region was studied (Figure 1) as relevant for the overall knowledge related to cattle genetic resources. Terreña (basque native breed) and other allochthonous breeds used in the region have been studied such as Salers, a rustic cattle breed used for beef production, which has grown in importance due to its ready adaptability to local management and environmental conditions. Furthermore, the Holstein-Friesian breed is primarily used as a cull dairy cow, which is also an integral part of the Basque regional beef supply chain. The exact test for population differentiation based on allele frequency showed that the breeds were significantly different from each other (*p* < 0.001). Pirenaica and terreña breeds are both native from the Basque region and showed certain admixture (Figure 1). A marked differentiation of Pirenaica was observed in the FCA plot in comparison with Salers and Holstein-Friesians (Figure 1a). These results confirm the genetic differences among native Pirenaica compared with Salers and Holstein-Friesian breeds, which have a distant geographical origin in Europe. The NJ phylogenetic tree showed three branches that mainly corresponded to Pirenaicas, Salers and Holstein-Friesians (Figure 1b). The Terreña breed was not well separated in the NJ tree, and even it has been less subjected to intensive selection. Balanced samples might be necessary for future studies of relationships among Basque native breeds. Clustering analysis using a Bayesian approach identified three underlying genetic clusters (Figure 1c). According to the probabilities of K as a log-likelihood given K clusters [32], the corresponding ΔK statistic showed that maximal ΔK occurred at K = 2. Pirenaica and Terreña breeds presented the zebu-diagnostic BM2113-131 allele, which has also been described in Salers [16]. Therefore, the ΔK method may underestimate K considering a common African genetic signature in these three breeds, whereas the geographical origin may indicate an optimal K = 3. Finally, genetic differentiation among breeds was considered. In our case, the overall genetic differentiation among breeds (F_ST_) was high (0.1195; Table 2), as it affects the performance of assignment tests. Genetically divergent breeds (F_ST_ > 0.1) are more likely to be correctly assigned than closely related ones (F_ST_ < 0.05) [6].

### 3.2. Individual Identification and Parentage Determination

In Pirenaica, the CMP value was 1.89 × 10^−25^ when 30 markers were used. Whereas, ISAG core panel of 12 STRs available in Genotype Panel 1.2 (Thermo Scientific) showed a CMP value of 2.3 × 10^−11^. The CMP value was 3.4 × 10^−13^ when the most polymorphic (highest PIC value) 12 markers were selected. The present study, using 12 STR core panel or using the most polymorphic 12 STRs, showed stronger discrimination power than previous studies [44,45]. Bovine Genotype Panel 2.2 (Thermo Scientific), which includes six additional STRs, could not be completely evaluated since three of its STRs (MGTG4B, RM67 and SPS113) were not studied following the recommendations [10]. However, laboratories that perform bovine parentage analyses usually use other complementary panels when more discrimination power is required for resolving complicated parentage cases.

In paternity testing, PE for each locus was measured as the probability of excluding an individual (sire) from being the father of the calf. PE was computed considering one known parent (PE-1P, dam not typed and random sire matched against calf) or two known parents (PE-2P, random sire matched against dam/calf pairs). In Pirenaica, the CPE for PE-1P and PE-2P was computed considering all 30 loci, but also considering the reduced loci number used in reference studies. Pirenaica showed similar CPE_1_ (0.9911; Figure 2) to Terreña (0.9918) when 11 markers were used [15], while CPE_2_ was 0.9997 in both breeds. When using the minimum 12-STR core panel recognized by ISAG, Pirenaica had a CPE_1_ and CPE_2_ value of 0.9946 and 0.9998, respectively. In other European breeds, 12-STR panel showed CPE_1_ values that ranged from 0.9135 to 0.9777 and CPE_2_ values that ranged from 0.9935 to 0.9999 [11]. However, a CPE value over 99.99% is necessary for paternity analysis, and, therefore, the 12-STR panel might be insufficient depending on breed and parentage. This evidenced the need to increase the number of STR loci in Pirenaica, in order to have enough exclusion power to resolve satisfactorily parentage cases. Van de Goor [11] also considered a 16-STR panel in several European bovine breeds showing a CPE_1_ values ranging from 0.9818 to 0.9994, and a CPE_2_ values from 0.9935 and 0.9999. However, the Pirenaica was not included in their forensic study.

The recommended 30-STR panel, used in this study, showed a CPE_1_ (considering one known parent) and CPE_2_ (two known parents) values of 0.99998 and 0.99999997, respectively. Our Pirenaica results showed that a 21-STR panel might be necessary to obtain a CPE_1_ value of 0.9999, while a 13-STR panel is enough for a discriminative CPE_2_ value of 0.9999 (Figure 2). Finally, for a sibling analysis in the Pirenaica breed, a 11-STR panel is enough for a CPE_SI_ value of 0.9999, which is similar to the values observed in other European breeds [11]. Overall, for most forensic cases, except CPE_SI_, the 12-STR core panel seems insufficient in Pirenaica and an increase in the amount of genotyped STRs should be considered. A 21-STR panel looks more appropriate to resolve some parentage analysis in Pirenaica.

### 3.3. Assignment Analysis of Breeds

In the assignments of geneclass, over 99% of individuals were allocated within their populations using both frequency (99.67%) and Bayesian (99.70%) methods using 30 STRs. Previous studies showed an assignment success between 67% and 100% using 6 to 23 STR data from 4 to 7 cattle breeds [6,44]. However, the assignment method and the confidence of the test should be considered since they greatly influence the assignment success. Genetic differentiation among breeds is also of interest as it affects the performance of assignment tests; genetically divergent breeds (F_ST_ > 0.1) are more likely to be correctly assigned than closely related ones (F_ST_ < 0.05) [6]. In our case, the mean F_ST_ among breeds was also slightly higher (Table 2) than the aforementioned studies. Only low differentiation in allele frequencies was observed between Pirenaica and Terreña native breeds (F_ST_ < 0.1).

In the Bayesian assignment of structure, a study, using the same 30 STRs recommended by ISAG-FAO, showed an assignment between 89.3% and 95.8% in Korean native cattle [46]. In this study, considering the 30 STRs panel and a cutoff value of 80% (Q ≥ 0.8), the overall proportion of animals correctly assigned to a breed was 98.8%. However, assignment tests were performed using a 11 STR panel and Basque native Terreña genetic data [15]; Pirenaica assignment decreased to 66.1%, whereas 31% of Pirenaica animals were uncorrected assigned to Terreña breed. In contrast, the Terreña, Salers and Holstein-Friesian showed higher assignments, 85.3, 94.8 and 96.6%, respectively. Therefore, a structure assignment test suggests that the number of STRs should be increased when other native breeds are included. Although previous methods (geneclass or structure) have been extensively used for the assignment, they encounter several limitations. The frequency-based method lacks the *p* value for measuring the confidence with which an individual belongs to a given population. Whereas, previous Bayesian methods can bias assignments or provide inaccurate results if sample sizes are unbalanced among populations [35,47]. In order to overcome the problem of unbalanced population sizes, a machine-learning approach (assignPOP package, r software) was applied to study the mean and variance of assignments (Figure 3). This approach combines various markers panels into a single predictive model, not possible in previous methods, while it can provide the minimum number and most discriminative markers necessary for accurate assignments. Firstly, Pirenaica was assessed against the 11-STR panel used in the Terreña breed [15] (Figure 3a). The means of assignment were generally low in Pirenaica (62.9%) and Terreña (58.2%), while high variance was observed in the bar-plots due to the non-accuracy of the assignment tests. In general, the 11-STR panel might not be enough for an accurate assessment of an individual to its population of origin, when geographically and phylogenetically related native Pirenaica and Terreña breeds are studied. A 16-STR panel reported in Blonde d´Aquitaine and Limousin [11], breeds also included in the Basque PGI label, was used against Pirenaica (Figure 3b). High mean assignment value was observed in Pirenaica (98.9%) in comparison to Blonde d’Aquitaine (45.4%) and Limousin (54.1%) considering balanced training sets of 25, 50 and 75 individuals. The overall mean assignment was of 66.1%, which increases with the amount of STRs included. In our study, lower assignments in Blonde d’Aquitaine and Limousin should be carefully considered since kinship is unknown in these breeds’ data. These two breeds are native from the south of France, a close Basque region where they have also been traditionally raised, and therefore they are included in the local PGI. However, future sampling and genotyping of Blonde and Limousin grown in the Basque region could obtain more trustable assignments along with Pirenaica.

Finally, several subpanels (11, 16 and 21 STRs) and a 30-STR panel assignment were compared (Figure 3c). Pirenaica and Salers showed over 90% mean assignments for the 16, 21 and 30-STR panels, while percentages were lower for the 11-STR panel with 70.4% and 83.5%, respectively. In contrast, for Holstein-Friesians a 21-STR panel was required to reach an assignment of 90% probably related to a reduced genetic diversity promoted by its selection for milk production. In general, 11 or 12-STR commercial panels seem to be insufficient for reliable assignment tests even selecting highest F_ST_ markers. Pirenaica showed that a panel of over 21-STRs is necessary for trustworthy assignments (≥95%; Appendix A). In essence, a reliable molecular traceability to ensure a correct assessment of an unknown beef product to its origin PGI label will depend on a well-designed STR panel containing the minimum amount of STRs with the highest F_ST_ values.

Up to date parentage control in cattle has been mainly based on STR but it is currently moving towards single nucleotide polymorphism (SNP)-based methods [48]. STRs are highly polymorphic and spread throughout the entire genome [49]. However, its analyses are time consuming, even for trained staff, due to the inconsistencies in allele size calling and errors in size determination by different laboratories. On the other hand, even SNPs are biallelic markers, high-throughput sequencing has permitted the development of high-density SNP panels with sufficient power to uniquely identify individuals. These SNP panels have increasing advantages such as greater abundance, genetic stability, simpler nomenclature and manageable to automated analysis [50]. In fact, a core panel of 100 and 100 additional SNPs have been defined for parentage control [51]. The cost per SNP is low compared to microsatellites, but the cost of the high-density assays might be prohibitive for many applications and the equipment necessary for high-throughput SNP panels is still quite expensive [10]. In contrast, in the present study, it is estimated a cost of EUR 10–20/sample for multiplex STR genotyping is needed, which makes this approach affordable for routine traceability in the food supply chain. Moreover, in case of the Pirenaica breed, the heterozygosity and the availability of large databases of STRs in cattle herd books, supposes a considerable reason to keep using these polymorphisms although SNPs will also provide promising advantages that need further research to evaluate the traceability and forensic effectiveness of an SNP panel in the Pirenaica breed.

## 4. Conclusions

The population data presented in this study provide extensive information regarding the discrimination power of a 12-STR ISAG core panel compared with the 30-STR complete panel. It has been demonstrated that the 30-STR panel is necessary as a first step to select the most appropriate markers in Pirenaica breed. In fact, the 21-STR panel is necessary for paternity analyses reaching in most cases a CPE_1_ value over 99.99%. Assignment tests using subpanels and some commercial panels are insufficient and the number of markers should also be increased for reliable assignments in Pirenaica and other breeds typically used for beef production in the Basque region. Overall, the present study recommends a new assignment approach by selecting the most polymorphic markers in order to design an appropriate STR panel to increase its discrimination power at a reduced time and cost of analysis.

## Figures and Tables

**Figure 1 animals-10-01584-f001:**
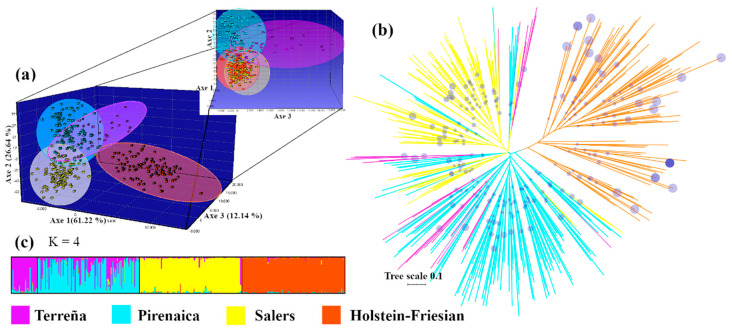
Genetic variations and phylogenetic differentiation analyses among cattle breeds. (**a**) 3D-FCA, (**b**) NJ radiation tree (Scale measured in R_ST_ distance values), size of circles represents bootstraps percentage and (**c**) mean probabilities of individual cluster memberships using structure (K = 4).

**Figure 2 animals-10-01584-f002:**
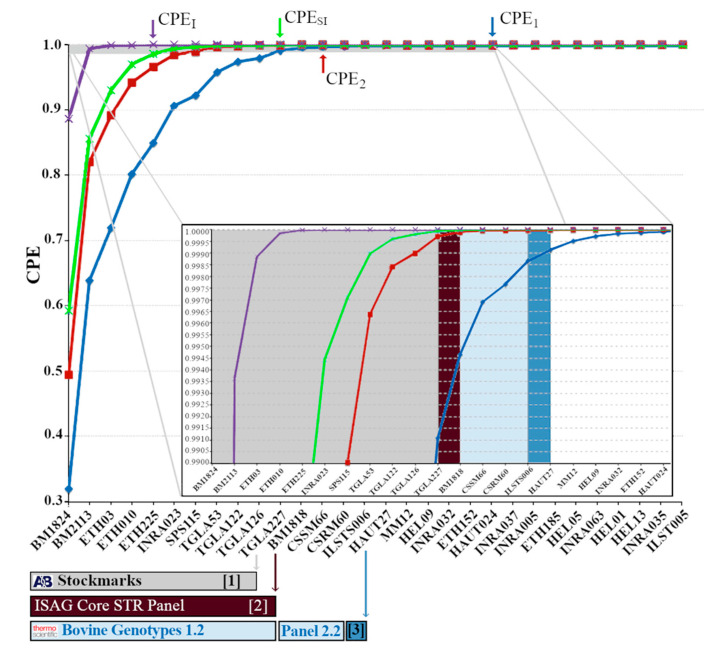
Comparison of combined power of exclusions (CPE) between the panels of short tandem repeat markers. Short tandem repeats (STRs) considering one known parent, two known parents, sibling and identical are shown in blue (CPE_1_), red (CPE_2_), green (CPE_SI_), and purple (CPE_I_ = 1 − CMP) cases, respectively. Little arrows (top) show the minimal number of markers for 99.99% of CPE. Boxes (below) show the markers included in recommended and commercial STR panels (grey, Stockmarks; garnet, ISAG core; blue, Bovine genotype 1.2 and 2.2). Numbers in brackets shows other panels used in literature.

**Figure 3 animals-10-01584-f003:**
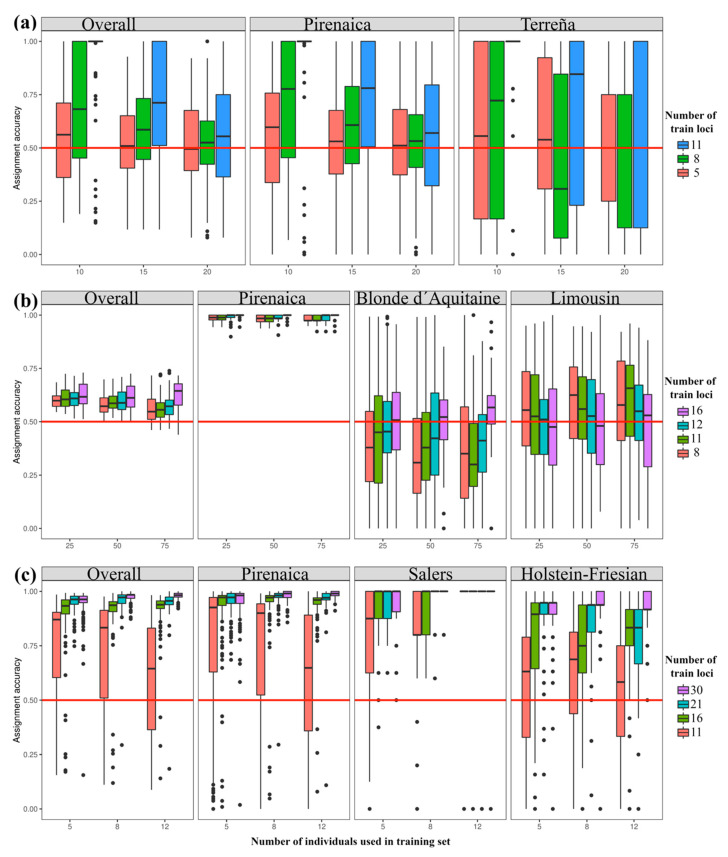
Assignment accuracies (%), according to Monte-Carlo cross-validation and depicted as bar-plots. (**a**) Pirenaica and Terreña breeds’ assignments for balanced populations (10, 15 and 20 individuals) crossed by three levels of train STRs with the highest F_ST_ (red: 5-STR; green: 8-STR) and all loci (blue: 11-STR), (**b**) Pirenaica, Blonde d´Aquitaine and Limousin breeds’ assignments for balanced populations (25, 50 and 75 individuals) crossed by four levels of train STRs with the highest F_ST_ (red: 8-STR; green: 11-STR; turquoise: 12-STR) and all loci (fuchsia: 16-STR). (**c**) Pirenaica, Salers and Holstein-Friesian breeds’ assignments for balanced populations (5, 8 and 12 individuals) crossed by four levels of train STRs with the highest F_ST_ (red: 11-STR; green: 16-STR; turquoise: 21-STR) and all loci (fuchsia: 30-STR).

**Table 1 animals-10-01584-t001:** Statistical parameters for genetic characteristics of Pirenaica breed using the panel of 30 short tandem repeat markers by ISAG-FAO [10].

Marker	k	Ho	He	PIC	MP	PE-1P	PE-2P	PE-SI	HW
**BM1824**	4	0.711	0.745	0.694	0.114	0.319	0.668	0.592	NS
**BM2113**	8	0.816	0.824	0.797	0.062	0.469	0.821	0.646	NS
**ETH03**	8	0.588	0.621	0.582	0.181	0.224	0.597	0.513	NS
**ETH010**	6	0.746	0.710	0.658	0.160	0.295	0.649	0.570	NS
**ETH225**	6	0.684	0.639	0.604	0.161	0.241	0.623	0.527	NS
**INRA023**	7	0.737	0.769	0.729	0.088	0.373	0.736	0.610	NS
**SPS115**	6	0.596	0.581	0.522	0.235	0.182	0.503	0.481	NS
**TGLA53**	11	0.754	0.820	0.791	0.063	0.462	0.815	0.643	NS
**TGLA122**	11	0.746	0.768	0.734	0.084	0.387	0.758	0.611	NS
**TGLA126**	5	0.658	0.608	0.556	0.201	0.203	0.545	0.502	NS
**TGLA227**	11	0.860	0.865	0.845	0.039	0.559	0.882	0.672	NS
**BM1818**	7	0.728	0.783	0.750	0.080	0.401	0.770	0.620	NS
**CSSM66**	12	0.732	0.777	0.755	0.078	0.422	0.808	0.619	NS
**CSRM60**	7	0.652	0.632	0.602	0.163	0.240	0.633	0.523	NS
**ILSTS006**	7	0.719	0.806	0.774	0.070	0.432	0.789	0.634	NS
**HAUT27**	7	0.693	0.745	0.702	0.111	0.342	0.706	0.594	NS
**MM12**	11	0.842	0.804	0.777	0.086	0.445	0.810	0.634	NS
**HEL09**	7	0.754	0.798	0.764	0.075	0.417	0.776	0.629	NS
**INRA032**	8	0.693	0.747	0.710	0.099	0.351	0.725	0.597	NS
**ETH152**	6	0.788	0.746	0.701	0.124	0.336	0.696	0.594	NS
**HAUT024**	8	0.772	0.737	0.691	0.123	0.327	0.687	0.589	NS
**INRA037**	9	0.268	0.663	0.601	0.231	0.250	0.588	0.536	***
**INRA005**	4	0.604	0.642	0.564	0.198	0.206	0.500	0.518	NS
**ETH185**	8	0.536	0.596	0.554	0.222	0.202	0.563	0.496	NS
**HEL05**	8	0.561	0.567	0.535	0.217	0.185	0.555	0.477	NS
**INRA063**	4	0.563	0.576	0.506	0.243	0.172	0.469	0.474	NS
**HEL01**	5	0.554	0.558	0.505	0.254	0.165	0.486	0.466	NS
**HEL13**	5	0.482	0.469	0.421	0.352	0.113	0.400	0.401	NS
**INRA035**	3	0.298	0.427	0.384	0.397	0.091	0.357	0.370	NS
**ILST005**	2	0.351	0.372	0.302	0.463	0.069	0.237	0.319	NS

k: number of alleles per locus; Ho: observed homozygosity; He: expected heterozygosity; PIC: polymorphic information content; MP: matching probability; PE-1P: power of exclusion for 1 known parent; PE-2P: power of exclusion for 2 known parents; PE-SI: power of exclusion for sibling; HW: Hardy–Weinberg equilibrium; NS: non-significant; ***, *p* < 0.001.

**Table 2 animals-10-01584-t002:** Pairwise F_ST_ (below diagonal) and R_ST_ (above diagonal) between breeds of cattle.

	Terreña	Pirenaica	Salers	Holstein-Friesian
Terreña	-	0.04277	0.14579	0.14030
Pirenaica	0.04102	-	0.11512	0.14115
Salers	0.12724	0.10324	-	0.24822
Holstein-Friesian	0.12304	0.12369	0.19886	-

All F_ST_ and R_ST_ values are significant (*p* < 0.001).

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
