# Peer review of "Genetic Characterization of the Local Pirenaica Cattle for Parentage and Traceability Purposes"

_animals, 2020, doi:10.3390/ani10091584_

Round 1

Reviewer 1 Report

This study by Gamarra et al aimed to assess the effectiveness of a 30-STR panel and explore the genetic structure of local Pirenaica breed compared with other breeds raised in the same geographic Basque Region. Their finding suggested the proposed STR panel could be used as an appropriate genetic tool to trace Pirenaica animals and their PGI products. Furthermore, they revealed a 21-STR panel with the highest FST markers is proposed for a confident parentage analysis and high traceability. General, the manuscript is interest and well designed, especially for genetic characterization of the local Pirenaica cattle.

  1. Line 76, how to collect the Muscle samples from Pirenaica beef cattle? these Muscle samples were collected after slaughter?
  2. Line 110, what is difference between “factorial components analysis (FCA)”and “PCA”? why not use PCA analysis?
  3. Line 155, please offer website for ISAG/FAO to obtain information about STR loci.
  4. Line 160, In the phylogenetic relationships section, the author compared Pirenaica with Salers and Holstein-Friesian, it is quite different from local breeds. Please clarify why these two breed was used for comparison.
  5. Line 193, “on allele frequency variations”> “on allele frequency”
  6. Line 195, “in the FCA plot (Figure 2a).”> “ in the FCA plot (Figure 1a).”;
  7. Line 198, (Figure 2b).? also, (Figure 1c) and (Figure 1b) need to added in the main text.
  8. Line 250, why the Figure 1d was presented here? The authors should clearly check the all tables across manuscript.
  9. In lines 318~328, the author compares the cost of multiplex STR genotyping and high-throughput SNP panels. The advantages and disadvantages of these two methods should also be discussed in more detail.
  10. There are some spelling errors in the manuscript, such as line 27, 1.87 x 10-25. Please check across whole  manuscript.
  11. Line 230, “However, Pirenaica was not included in their forensic study.、”, remove “、”.
  12. Figure 1 a), b), while figure 3(a), 3(b), Figure legends should be consistent.

Reviewer 2 Report

The main aim of the manuscript named “Genetic characterization of the local Pirenaica cattle for parentage and traceability purposes” was to evaluate the effectiveness of a 30-STR panel for study the genetic structure of local Pirenaica breed compared with other breeds raised in the same geographic Basque Region and trace Pirenaica animals and their PGI products.

The present work has a moderate level of originality, however, the results obtained could be used to conserve the Pirenaica native breed and to guarantee the its PGI certification. However, before this manuscript can be approved for publication in the journal Animals, it is recommended to take into account the following points:

Lines 45 and 46. Replace the references by their corresponding numbers.

Line 56. A new version of the commercial kit was recently released that include more STRs.

Lines 83-92. It is not necessary to describe the extraction method, only mention it.

Line 108. “Phylogenetic relationships” could be replaced by “genetic relationships” because Phylogenetic implies evolutionary relationships between the breed analyzed that is not exactly right for bovine breeds.

Lines 111 to 116. These lines could be moved to the Assignment test paragraph.

Lines 119 to 122. CMP and PE-I are complementary, consider to  use only one of these parameters.

Lines 123 to 129. The inbreeding values could be estimated and include in the PE formulas.

Lines 141. Structure results could also use for comparison.

Lines 188 to 190. Population reduction affects the number of alleles more than heterozygosity by eliminating rare alleles. What happens when the number of alleles is compared?

Lines 191 to 202. There data available from other native Basque breeds to include in these analyses?

Line 213 to 215. The new commercial kit includes 18 STR markers, the 12 ISAG core panel more 6 additional microsatellites. In addition, lab that perform bovine parentage analyses usually use complementary panels.

Line 262. Replace the reference by its corresponded number.

The results showed in the section “3.3. Assignment analysis of breeds” could be compared with some of the previous works that evaluate assignment test in bovine using STR panels.

The section “3. Results” should be “3. Results and Discussion” thus the next section is “conclusions”.

Author Response

Response to Reviewer 2 Comments

The main aim of the manuscript named “Genetic characterization of the local Pirenaica cattle for parentage and traceability purposes” was to evaluate the effectiveness of a 30-STR panel for study the genetic structure of local Pirenaica breed compared with other breeds raised in the same geographic Basque Region and trace Pirenaica animals and their PGI products.

The present work has a moderate level of originality, however, the results obtained could be used to conserve the Pirenaica native breed and to guarantee the its PGI certification. However, before this manuscript can be approved for publication in the journal Animals, it is recommended to take into account the following points:

  1. Lines 45 and 46. Replace the references by their corresponding numbers.

Response 1: The format of references have been corrected.

  1. Line 56. A new version of the commercial kit was recently released that include more STRs.

Response 2: A new comment has been written according to reviewer´s suggestion including the new comercial kit that includes more STRs. (L59-L60).

  1. Lines 83-92. It is not necessary to describe the extraction method, only mention it.

Response 3: Extraction method has been only mentioned as proposed by reviewer (L88).

  1. Line 108. “Phylogenetic relationships” could be replaced by “genetic relationships” because Phylogenetic implies evolutionary relationships between the breed analyzed that is not exactly right for bovine breeds.

Response 4: Following the recommendation of reviewer, “Genetic relationships” has been written in the manuscript (L187).

  1. Lines 111 to 116. These lines could be moved to the Assignment test paragraph.

Response 5: Following the recommendation of the reviewer, the description of structure method have been moved to the assignment test paragraph (L144-L149).

  1. Lines 119 to 122. CMP and PE-I are complementary, consider to  use only one of these parameters.

Response 6: PEI has been eliminated from Material & Methods section and Table 1. The CMP has been used in the results and discussion, so it´s values could be compared with previous studies. (L254-258).

  1. Lines 123 to 129. The inbreeding values could be estimated and include in the PE formulas.

Response 7: We appreciate the suggestion of the reviewer. However, we have not been able to find an appropiate reference that allow us to estimate new PE formulas including inbreeding (?). Generally, we applied theta computed from population to solve specific (trios or duos) parentage cases, but not for in PE adjustment. Therefore, we would appreciate to receive a literature reference or other suggestion from reviewer, so we can improve this part of the manuscript.

  1. Lines 141. Structure results could also use for comparison.

Response 8: Suggestion of reviewer has been considered. Therefore, Structure has been commented in Material and Methods section (L144-L149). These results were shown (before revision) in the 3. Results and discussion (3.1. Genetic variations and genetic relationships section (L318-326).

However, the assignment results of structure have been moved to 3. Results and discussion (3.3. Assignment analysis of breeds), so they can be compared with other methods of assignment (L311-L326).

  1. Lines 188 to 190. Population reduction affects the number of alleles more than heterozygosity by eliminating rare alleles. What happens when the number of alleles is compared?

Response 9: Suggestion of reviewer has been considered and the number of alleles per locus in Pirenaica has been commented in comparison with other native breeds from previous studies. (L230-233).

  1. Lines 191 to 202. There data available from other native Basque breeds to include in these analyses?

Response 10:   We appreciate the suggestion of reviewer.

Therefore, genetic data from Terreña (native basque breed) has been used to analyze genetic relationships. We have constructed a new Figure 1 [(a) new 3D-plots; (b) NJ tree and (c) Structure analysis] showing genetic relationships among 4 breeds (Pirenaica, Terreña, Salers and Holstein-Friesian). Instead of Figure 1d (before revision), a Table 2 is now shown (genetic distance matrix) which is much convenient to show data when 4 populations are used. Results and discussion has been modified according to the new figure (L223-242).

  1. Line 213 to 215. The new commercial kit includes 18 STR markers, the 12 ISAG core panel more 6 additional microsatellites. In addition, lab that perform bovine parentage analyses usually use complementary panels.

Response 11: The paragraph has been modified following reviewer´s suggestion to consider other analyses in which other STRs are included to improve discriminative power. (L273-L275)

  1. Line 262. Replace the reference by its corresponded number.

Response 12: The format of references have been corrected.

  1. The results showed in the section “3.3. Assignment analysis of breeds” could be compared with some of the previous works that evaluate assignment test in bovine using STR panels.

Response 13: Following the suggestion of reviewer, different assignment methods used in the present study has been compared with some previous studies (L298-L307). In addition, other basque native breed has been included (Response: 10), therefore discussion assignment analysis have been accordingly modified.

  1. The section “3. Results” should be “3. Results and Discussion” thus the next section is “conclusions”.

Response 14: The section has been renamed to 3. Results and Discussion.